# A Comparative Study of Pickled Salted Eggs by Positive and Negative Pressure-Ultrasonic Method

**DOI:** 10.3390/foods12071477

**Published:** 2023-03-31

**Authors:** Chaogeng Xiao, Yue Zhang, Ting Gong, Wenjing Lu, Di Chen, Cen Zhang, Haiyan Wang, Rongfa Guan

**Affiliations:** 1State Key Laboratory for Managing Biotic and Chemical Threats to the Quality and Safety of Agro-Products, Institute of Food Science, Zhejiang Academy of Agricultural Sciences, 298 Desheng Road, Hangzhou 310021, China; 2College of Life Sciences, China Jiliang University, Hangzhou 310021, China; 3College of Food Science and Technology, Zhejiang University of Technology, Huzhou 310014, China

**Keywords:** salted egg, flavor, ultrasonic, positive and negative pressure

## Abstract

In this study, the positive and negative pressure-ultrasonic method was applied to salted egg pickling, compared with traditional pickled salted eggs by various physical and chemical indicators. Results indicated the salt content of egg white and egg yolk increased rapidly in the salt-preserved salted egg with the positive and negative pressure-ultrasonic method, and the moisture content decreased rapidly. In addition, the oil yield of egg yolk was marinated for 12 days compared with the normal method of 35 days, and the ripening time of salted eggs was shortened by 2/3. There was no obvious difference in the microscopical structure of the egg yolk between the two methods of pickling. Moreover, the pores on the eggshell of the salted egg that was marinated by the positive and negative pressure-ultrasonic method had big cracks, which was beneficial to the substance exchange of the eggs and the outside. The common volatile flavor substances were detected by GC-MS, and a total of 33 flavor constituents were detected. There was no significant difference between the content of alcohols, aldehydes, and ketones, which contributed greatly to the flavor. Overall, the results indicated that this innovative salted eggs method can significantly reduce the curing time while ensuring the quality of salted eggs.

## 1. Introduction

In Asian nations, salted eggs are one of the most traditional preserved egg products [1,2,3]. Fresh duck eggs are usually made into a variety of processed eggs because of their heavy astringency, while duck eggs could be preserved by salting to keep their quality and lengthen their storage time [4,5,6]. Salted duck eggs have a long history and are very popular among people in China [1]. The salted yolks are served as stuffing material in several delicacies such as moon cake, other pastries, and glutinous rice dumplings, in addition to serving as a normal diet in the form of whole eggs [7,8]. The salted duck egg is rich in nutrients, containing abundant fat, protein, various amino acids, calcium, phosphorus, iron, various trace elements, and vitamins, which are absorbed by the body, has a moderate salty taste, and is suitable for both young and old consumers [1].

At present, the conventional processing method for the production of salted eggs is the brine immersion test, which is a simple process with low production costs and is used by enterprises for mass production [9]. However, the curing time is too long; it usually takes 25 to 35 days in summer and 50 days in winter to make the salted eggs [9,10,11,12]. Factors such as excessive bacteria and salty black spots have always been the key to research.

Researchers have found that it is possible to change the transfer rate of material inside and outside the eggshell by changing pressure and using the ultrasonic wave, which resulted in accelerating the curing efficiency [13,14]. At present, salted eggs are usually marinated by ultrasonic waves or pulsating pressures, respectively [13]. The pickling rate is not high, and the ripening time of salted eggs is longer. The positive and negative pressure phase of ultrasound can produce a beneficial cavitation effect, which will have a positive effect in the salted egg curing process [15,16]. Lin [17] found that the cavitation of ultrasonic waves can reduce the viscosity of egg white, and a reasonable ultrasonic treatment can promote the diffusion of sodium chloride molecules. Fan [13] showed that the repeated use of ultrasound can not only promote the penetration of salt but also improve the eating quality of salted eggs. Wang [18] found that the effect of ultrasound was better in the first few days before pickling. Yin [19] used positive and negative pressure technology to marinate duck meat. The experimental results showed that the salt penetration rate of duck meat was significantly higher than that of the control group. Moreover, salted eggs under negative pressure conditions can reduce the number of days of pickling and effectively inhibit microbial growth [20,21,22]. Previous studies found that the pulsation ratio of high pressure and normal pressure was 10 times higher than that of high-pressure continuous curing by studying the characteristics of eggshells [18,23].

In this study, duck eggs were used as research materials to study the effectiveness of different pickling methods. The combination of ultrasonic pretreatment and positive- and negative-pressure pickling was applied to the salted egg pickling process to reduce the curing time and improve the curing efficiency and the quality of the pickled products. The differences between the positive and negative pressure-ultrasonic curing method and the conventional pickling method were compared to provide a new research idea for the pickling of salted eggs.

## 2. Materials and Methods

### 2.1. Materials and Instruments

Fresh duck eggs with uniform size, oval shape, and weighing about 70 g were bought from Zhejiang Hangzhou Shenshen farmer’s market. Silver nitrate, potassium chromate, glutaraldehyde, phosphoric acid, osmium tetroxide, ethanol, and isoamyl acetate were purchased from the analysis of the pure Chinese medicine group.

The rapid pickling device (Appendix A) was in the Institute of Food Science, Zhejiang Academy of Agricultural Sciences.

### 2.2. Traditional Pickling Method

This method was slightly modified by Kaewmanee [24]. Each duck egg was immersed separately in a brine solution (25%, *w*/*v*). Both groups of duck eggs were stored at room temperature (28 to 30 °C) and removed weekly during the curing period up to 4 weeks.

### 2.3. Positive and Negative Pressure-Ultrasonic Marinated Salted Eggs

Duck eggs of similar size and without damage were selected, cleaned, and dried into pickling containers, poured with marinade, and sealed, and pickling parameters were set (high-pressure amplitude of 120 kPa, low-pressure amplitude of −70 kPa, positive- and negative-pressure time ratio of 16:24 min, ultrasonic effect in the first three days of pickling, and daily action time of 30 min) for sampling tests.

### 2.4. Determination of Salt Content

The sodium chloride concentration in egg samples was measured by the method of AOAC (2000) using direct titration. The direct titration method was used to measure the sodium chloride content of salted eggs. We weighed *m* g of analytically pure NaCl, accurate to 0.0002 g, added to a 250 mL conical flask, dissolved with about 70 mL of water, and added 1 mL of 5% potassium chromate solution. It was then shaken evenly, and we recorded the volume of AgNO_3_ standard titration solution consumed when the drop was red-yellow (kept for 1 min without fading) (*V*_2_). The exact value of the solution was calculate d(*c*_2_) as:(1)$$c_{0}=\frac{m}{0.0584\times V_{2}}$$

Next, 25 g (accurate to 0.001 g) of pounded sample (recorded as *m*) was taken in a 250 mL conical flask, and 100 mL of hot water was added at 70 °C; it was then shaken well, cooled, and filtered, and the volume was fixed to 250 mL, to be measured.

Then, 10 mL of filtrate, 50 mL of water, and 1 mL of 5% potassium chromate solution were added to the 250 mL conical flask. The volume of 0.1 mol/L AgNO_3_ solution consumed was recorded when the solution was dropped to red-yellow (kept for 1 min without fading) (recorded as *V*_1_), and the volume of AgNO_3_ consumed in the blank test was recorded (recorded as *V*_0_).

### 2.5. Determination of Egg Yolk Oil Yield

Oil exudation of egg samples was measured according to the method of Lai [25] with a slight modification. A total of 3 g of a fresh mixed egg was added with 21 mL of n-hexane and 14 mL of isopropanol. The mixture was stirred evenly with a magnetic stirrer, and then it was concentrated and evaporated in a rotary evaporator at 55 °C, and at last, was dried in a drying oven at 105 °C until constant weight. The filter residue was weighed as the total lipid content. A total of 3 g sample was added with 25 mL of distilled water, homogenized for 30 s, and centrifuged at 1000 r min^−1^ for 30 min. The suspension was added with 15 mL of n-hexane and 10 mL of isopropanol to dissolve the lipid layer. The lipid layer was also concentrated and evaporated in a rotary evaporator at 55 °C, then dried in a drying oven at 105 °C to a constant weight and weighed to a free lipid mass.

### 2.6. Salted Egg Profile Texture Analysis

This method was slightly modified according to the method of Kaewmanee [8]. The P 36R probe was used in TPA mode (TA.XT.Plus, Stable Micro System, Godalming, UK) for texture determination, and the test parameters included hardness, springiness, chewiness, resilience, cohesiveness, and gumminess. Before testing, the samples were pretreated by cutting the cooked egg white into 1 cm × 1 cm × 0.5 cm-size pieces and cutting the egg yolk in half into uniform hemispheres. The egg white samples were compressed to 70% of the original volume with the probe, and the yolk samples were compressed to 80% of the original volume with the probe. The test speed was 1 mm s^−1^ with a dwell time of 5 s. Each sample was tested 10 times.

In textural characterization, hardness represents the mechanical properties related to the force required to deform or penetrate the egg white or yolk. It reflects the bonding force within the food to retain its shape. The gumminess demonstrates the adhesion and cohesion forces. When the sample has high adhesion, it indicates that the gumminess is high and the substance is not conducive to peeling when sliced. In addition, the value of cohesiveness is the ability of the salted egg to resist damage and stay close together to keep the egg body in its original state when the probe is compressed downward. The higher the cohesiveness indicates the tighter the bond between the tissues inside the food. Springiness reflects the ability of the salted egg to deform when it is compressed by the probe and to recover its original state when the probe is withdrawn. The chewiness is related to the three qualitative indices of hardness, cohesiveness, and springiness , and is intended to simulate the magnitude of the work required by the human mouth to chew an object to a state where it can be swallowed. Resilience reflects the degree to which the elastic energy preserved by the deformation of the salted egg when it is squeezed by the probe returns at the same speed and pressure.

### 2.7. The Microscopic Structure of Salted Eggs

This method was modified according to the method of Wang [9]. The raw salted egg yolk was cut into 3 mm × 3 mm × 1 mm-volume pieces and then pretreated. The samples were placed in a 2.5% glutaraldehyde solution and removed overnight at 4 °C. Afterward, the samples were rinsed, poured off the glutaraldehyde fixation solution, and rinsed three times with 0.1 M phosphoric acid solution at pH 7.0 for 15 min each time, followed by refixation with 1% osmium solution for 1–2 h. After rinsing, the samples were dehydrated using 50%, 70%, 80%, 90%, and 95% ethanol, each concentration for 15 min, and finally, with 100% ethanol; the samples were then submerged in a mixture of ethanol and isoamyl acetate (*V/V* = 1/1) for 30 min and then in pure isoamyl acetate for 1–2 h. Finally, the samples were observed after coating. 

### 2.8. Analysis of Volatile Flavor Components

This method employed GC-MS to investigate the volatile flavor components in egg samples, which were measured according to the method of Wei [26] with a slight modification. A total of 6 g of salted egg yolk sample was placed in a 20-mL injection empty bottle with a rubber septum cap to heat at 50 °C for 30 min. The extraction handle was inserted into the headspace vial and fixed with an iron stand, and the extraction tip of the front end was extracted for 30 min. After the extraction, the extraction needle was retracted and the extraction handle was pulled out; the extraction handle was then inserted into the injection port, and the extraction head was pushed out for temperament sample analysis at 240 °C for 2 min GC-MS analysis.

Chromatography was performed on a 7890 B gas chromatography (Agilent, Santa Clara, CA, USA) equipped with a 5977 B mass selective detector. Separation was achieved using a DB-WAX capillary column (i.d., 60 m × 0.25 mm; film thickness, 0.5 μm). GC-MS temperatures were as follows: injector, 240 °C; no sample mode, column flow rate: 1 mL min^−1^; column, 220 °C with an increment of 3 °C min^−1^; ion source, 230 °C; and quadrupole temperature, 150 °C. The mass spectrometer was programmed under electron impact (EI) in a full-scan mode at m z^−1^ 33–500 with a scanning rate of 2 scans s^−1^.

### 2.9. Data Analysis

The experimental data were processed with Excel 2016, and the orthogonal experimental design and significance analysis were performed with SPSS 19.0 software. Data were expressed as mean value ± standard deviation (SD) of at least triplicates.

## 3. Results

### 3.1. NaCl Content

The change of salt content between positive and negative pressure-ultrasonic pickling and conventionally pickled egg yolk was shown in Figure 1A. The positive and negative pressure-ultrasonic salted egg yolk had a faster increase of salt content because the salt in the egg white penetrated the egg yolk in the late stage of pickling. The salted egg yolks marinated for 12 days had a high salt content of 1.73%, which is higher than the salted eggs marinated for 30 days by conventional methods. The salt content of the conventionally marinated egg yolk showed a steady growth trend from the first 0 to 6 days because the salt entered the egg yolk and was blocked by the egg white macromolecular protein gel during the natural infiltration process. The salt had not penetrated the egg yolk.

As shown in Figure 1B, salted eggs pickled by positive and negative pressure-ultrasonic group were marinated on the 12th day. The present results compared with the results of Kaewmanee’s [7] studies could confirm that the combined positive and negative pressure-ultrasonic acid washing method greatly contributed to the increase in the salt content of egg whites. In the conventional group, the egg white growth rate tended to be gentle in the egg white from 21 to 30 days. The osmotic pressure inside and outside the eggshell was not different, the rate of salt entering the eggshell during the natural curing process was reduced, and the pores on the eggshell were easily blocked by salt.

### 3.2. Water Content

Figure 2A,B showed the changes in water content of egg yolk and egg white during curing by the conventional method and the positive and negative pressure-ultrasonic method. With the penetration of salt, the water content of egg white and yolk gradually decreased. Comparing Figure 2A,B, it can be seen that the change of yolk moisture was more significant than that in egg white, with a decrease of 26–30% for the yolk and 7–10% for egg white. This was because the dehydration effect of salt separates the yolk fat from water, and a large amount of water leaks out of the yolk and is discharged out of the egg in turn. Similar behavior was also observed in other studies [9,27].

As can be seen in Figure 2A, the positive and negative pressure-ultrasonic group showed a large decreasing trend in yolk salt content, especially on days 9–12, when the rate of water decrease was high. In contrast, the conventional group showed a slow decreasing rate from day 0 to 18, and the decreasing rate increased from day 18 to 21. This is because the cavitation effect of ultrasound can reduce the springiness of egg white and refine the egg white protein gel, thus reducing the obstruction of salt penetration. The high pressure caused the salt to penetrate further into the yolk while the vacuum environment promoted the drainage of water from the yolk.

### 3.3. Egg Yolk Oil Yield

The oil exudation of egg yolk is one of the maturation characteristics of salted eggs, and consumers commonly believe it to be the standard for judging the quality of salted eggs [28,29,30]. After high-temperature ripening, an egg yolk progressively moving from a solid form to a gel state under the action of salt would have oiliness and grittiness [24]. Moreover, the oil yield of egg yolk can reflect the maturity of salted eggs, and the mature salted egg yolk has a fine sand feeling and a high oil yield [31]. Figure 3 depicted the change in egg yolk oil yield with pickling time under two curing methods. In the positive and negative pressure-ultrasonic group, the oil yield of egg yolk depicted a high growth trend from 6 to 12 days. This was because the salt penetrated the egg yolk to destroy the stable yolk system of the egg yolk, and the water from the egg yolk was lost, resulting in the formation of an oil droplet. The oil yield of egg yolk reached 52.12% by the positive and negative pressure-ultrasonic method on the 12th day, and the oil yield of egg yolk was 54.79% by the conventional method on the 30th day. This result was also consistent with the result of Liu [28] that oil exudation of egg yolk increased significantly in the early stage of curing (0–21 days), and a high oil yield was accompanied by the formation of egg yolk granulation [28].

### 3.4. Texture Analysis of Salted Eggs

The texture parameters of salted eggs, such as hardness, springiness, and cohesiveness, reflect the taste and maturity of salted eggs in one aspect and are also important indicators for consumer acceptance [8]. To investigate the effect of ultrasonic action on the textural properties of salted eggs, the process of positive and negative pressure-ultrasonic salted egg pickling was picked. The texture of the egg white and yolk was respectively measured every other day by removing the salted egg samples from the pickling containers.

Table 1 shows the changes in composition of egg white, which illustrated that the hardness of salted egg white decreased by 33.93% with the increase in curing time compared to the egg white of fresh eggs. Additionally, the changes in hardness may be related to protein denaturation, where the high salt content promoted protein leaching and makes the cooked salted eggs hard [24,32], and also to the decrease in water content, where the hardness of salted eggs decreased with the decrease in water content until the water content was reduced to a certain level [33]. Springiness did not show significant difference, indicating the elasticity of egg white was unchanged. As for cohesiveness, because the value itself was small, the difference between the values was not significant. Gumminess and chewiness were reduced by 34.25% and 29.76%, respectively, due to the decrease in hardness. Gumminess was numerically equal to the product of hardness size, cohesiveness, and springiness. All values of hardness, cohesiveness, springiness, and gumminess was reduced, because the effect of salt during the pickling process leads to changes in the texture of egg white. Considering the change of resilience, the egg white gel properties had a greater effect on its resilience, while NaCl decreased the degree of gel cross-linking, leading to a decrease in gel properties and a consequent decrease in the resilience of egg white. The results of the same trend are similar to the results of Yu’s study [34].

Moreover, Table 2 describes the changes in yolk texture with curing time. With the continuous penetration of salt into the yolk, the hardness of the salted egg yolk decreased by 21.96% and the springiness decreased by 46.98% compared to the fresh egg yolk, and the rich protein of eggs had a certain ability to resist external forces, and this resistance was expressed as the springiness of the egg [23]. The infiltration of salt, however, led to the destruction of the original emulsion system of the yolk and the aggregation of some proteins, which reduced the ability to resist external forces, thus the springiness was reduced. This finding was consistent with previous studies [35]. The cohesiveness, gumminess chewiness, and resilience were reduced by 30.26%, 33.67%, 63.55%, and 44.73%, respectively.

### 3.5. The Microstructure of Salted Eggshell

The microstructures of fresh duck eggshells, fast-cured salted eggshells, and traditional cured salted eggshells were observed using scanning electron microscopy, and the scanning results were shown in Figure 4.

Overall, an eggshell seemed to be a composite layer made up of solid and membrane sections (about 0.4–0.8 mm thick). However, within the solid part, there was room for more diversity [14]. The solid part consisted of a dense cuticle with a dense appearance. The structure of the eggshell membrane of fresh eggs (Figure 4A) was more compact than that of salted eggs (Figure 4B,C). As in Figure 4, Figure 4A shows the fresh duck eggshell at 800 times magnification, and the small round holes in the figure are the stomata on the shell as the channels for material exchange, and it can be seen that the stomata of the fresh eggshell were small, which was not conducive to the material exchange between the interior of the duck egg and the pickling solution. Figure 4B indicated the shell of a conventionally cured mature salted egg, and small cracks can be seen on the shell at a magnification of 500 times, which was the enlargement of the stomata on the shell surface during the curing process due to the difference in osmotic pressure, salt entry, and water discharge. Figure 4C depicted larger cracks on the eggshell surface after magnification 500 times. The reason for this may be the oscillation of the ultrasonic waves causing the expansion of the eggshell cracks, or it may be the use of positive and negative pressure on the eggshell surface, resulting in larger gaps on the surface of the eggshell of positive and negative pressure-ultrasonic pickled salted eggs. It was the cracks created on the eggshell surface that led to larger channels for material transfer and accelerated salt penetration.

### 3.6. Microstructure of Salted Egg Yolk

The results showed that fresh egg yolk had no obvious graininess and exhibited a low lipid overall structure (Figure 5A). However, there was no significant difference in the microstructure between the fast-cured salted egg yolk and the conventional salted egg yolk (Figure 5B,C). The curing effect of the conventional pickling for 30 days was similar to that of 12 days. Compared with fresh egg yolk, the mature salted egg yolk under an electron microscope showed a phenomenon of superposition of polyhedral particles. Since the salt penetrated the egg yolk during the curing process, which destroyed the stability of the egg yolk, the egg yolk was continuously dehydrated. The polyhedral particles were more closely connected so that the cooked salted egg yolk had a loose taste. This finding revealed that using a rapid procedure to pickle salted egg yolks might hasten the production of a gritty texture. Dehydration was more evident during pickling and preservation, and the granules were closer and smaller. After 48 h of pickling and 72 h of preservation, the salted egg had a granular texture, making it acceptable for boiling [9]. The similar results were observed in the research of Yu and Wang [14,34].

### 3.7. Volatile Flavor Components of Salted Egg Yolk

After salted eggs were cured with NaCl, the salt penetrated the eggs, leading to partial denaturation and hydrolysis of proteins, fats, and other substances, resulting in small molecule flavor substances such as amino acids, esters, aldehydes, and alcohols. Compared to raw duck eggs’ fishy taste, the salted duck eggs’ yolk had a unique and strong flavor. Extraction of flavor substances from salted egg yolks was studied to reveal the flavor variation of salted egg yolks compared with fresh egg yolks, and to find out the flavor differences between them when compared with conventional salted egg yolks. In this study, solid-phase microextraction (SPME) was used to analyze the flavor of egg yolk.

When using SPME to analyze the flavor of egg yolk, some impurity peaks did not belong to the flavor of egg yolk, and these impurity peaks might come from the extraction head or capillary column and other instruments, and the impurity peaks were removed from the spectrum. The impurities that appeared in the analysis of volatile components of salted egg yolk using the 50/30 μm extraction head were shown in Table 3.

The volatile components of the yolks of the three groups were analyzed, and the GC-MS ionograms were obtained. The results are shown in Table 4. A total of 50 volatile compounds were detected in the yolks of the three groups, and these compounds were classified into nine categories, namely alcohols, nitrogenous, aromatic, hydrocarbons, aldehydes, ketones, ethers, acids, lipids, and heterocycles. Among them are isoamyl alcohol, n-octanal, benzaldehyde, nonanal, dodecane, decane, heptane, toluene, 1-octen-3-ol, hexanal, and acetone; these compounds were common to all three groups of egg yolks.

The ionogram of volatile compounds in fresh egg yolk is shown in Figure 6; it can be seen that 27 volatile compounds were detected in fresh egg yolk, among which hydrocarbon species were the most abundant, amounting to 15 types. The highest content of several compounds and their percentages were: 2,2,4,6,6-pentamethyl-heptane accounted for 10.08%, hexanal accounted for 9.84%, and decane accounted for 5.33%. Although 15 hydrocarbons were detected, the content of each hydrocarbon was small and did not contribute much to the odor of fresh egg yolk. The content of hexanal was not high and thus had little effect on the flavor of fresh egg yolks. Limonene, a lemon fruit odor, was detected only in the fresh duck eggs. Alcohols, which had a strong influence on flavor, accounted for only 2.96%, and aldehydes and ketones for 14.25%. Overall, the fresh duck egg yolk had a wide variety of volatiles, with a low content of each compound.

Positive and negative pressure-ultrasonic pickled salted egg yolk volatile flavor substances were shown in Figure 7. Table 4 depicts the rapidly pickled salted egg yolk volatile compounds that were detected. In total, there were 29: alcohols accounted for 25.49%, aldehydes and ketones accounted for 38.44%, hydrocarbons accounted for 3.235%, aromatics accounted for 9.04%, heterocyclic accounted for 0.77%, ether 4.0%. Among them, hexanal had the highest content, accounting for 28.71%, ethanol accounted for 16.03%, and toluene accounted for 9.04%, and the content of these three compounds accounted for 1/2 of the total volatile compounds, indicating that hexanal, ethanol, and toluene contributed the most to the flavor of salted eggs. The main substance causing the fishy flavor of salted duck eggs was hexanal. The content of acetic acid and 2-pentanyl-furan was not high and did not contribute much to the flavor of salted egg yolk. The ether substances identified were most likely obtained by dehydration of alcohols.

From the analysis of the differences in volatile substances between rapidly cured and conventionally cured salted egg yolks in Table 4, 1-pentanol, 1-butanol, 1-octen-3-ol, isoamyl alcohol, ethanol, 2,3-butanediol, hexanal, pentanal, acetone, 2-heptanone, n-octanal, benzaldehyde, nonanal, trans-2-octenal, ethyl-even-al, decane, heptane, dodecane, 2-pentanyl-furan, toluene, acetic acid, and methoxy-phenyl-oxime were identified. In the two groups of volatiles identified, 2-pentanyl-furan was generated by the Melad reaction, and in the conventional curing group, 2-pentanyl-furan accounted for 3.50% and had a greater effect on the flavor of salted egg yolk. Alcohols and aldehydes and ketones, which contributed more to the flavor, accounted for 21.6% of alcohols and 40.09% of aldehydes and ketones in the conventional pickled salted egg yolk, and 25.49% of alcohols and 38.44% of aldehydes and ketones in the rapidly pickled salted egg yolk group, with little difference in the proportions, indicating that the flavors of both were similar.

Conventionally cured salted egg yolks had more hexanol, n-octanol, 2,3-butanediol, 3-octanone, butyl acetate, and ethyl 3-hydroxybutyrate and less αα-dimethyl-2-propanol, acetophenone, 2-pentanone, and dimethyl sulfide than fast-cured salted egg yolks. The reason for the difference may be that the traditional salted duck eggs were cured for 30 days, and the aldehydes had enough time to be converted into esters, and there were more alcohols and esters than fast-cured salted egg yolks, but the difference in compound content was small, so there was no significant difference in volatile compounds in the salted egg yolks, and the flavor of salted eggs cured by the two methods was similar, proving that quick curing could guarantee the flavor of salted eggs while shortening the curing time.

## 4. Conclusions

In this study, an innovative positive and negative pressure ultrasonic method was used to salt eggs and to investigate the differences between it and the traditional salting method in terms of salt content, oil yield, moisture, microstructure, and volatile flavor substances of salted eggs. There were no significant differences between the salted eggs cured by this innovative method for 12 days and those cured by the traditional method for 30 days in terms of physicochemical indexes such as salt content of egg white, salt content of egg yolk, moisture content, and microstructure, but there were larger pores and cracks on the shell surface of the eggs cured by the positive and negative-pressure ultrasonic method. The volatile flavor substances of the mature salted eggs cured by the two methods were detected by GC-MS, and a total of 23 common substances were detected. There was no significant difference in the proportion of alcohols, aldehydes, and ketones, which contributed more to the flavor, indicating that the method ensured the flavor quality of salted eggs based on significantly shorter salting time.

## Figures and Tables

**Figure 1 foods-12-01477-f001:**
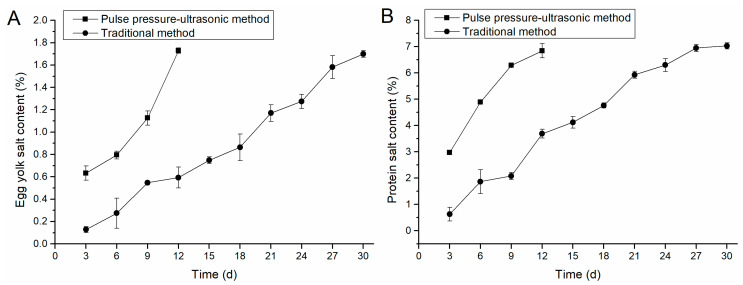
Comparison of salt content in egg yolk and egg white. (**A**) shows the variation of egg yolk salt content with time under different process conditions. (**B**) illustrates the variation of egg-white salt content with time under different process conditions.

**Figure 2 foods-12-01477-f002:**
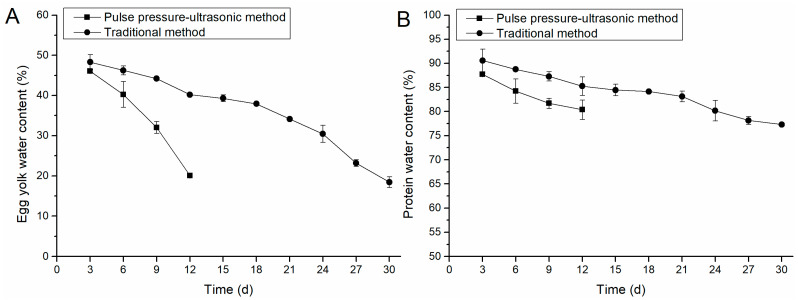
Comparison of moisture content in egg yolk and egg white. (**A**) illustrates the changes in egg yolk water content by pulse pressure-ultrasonic method and traditional method; (**B**) illustrates the changes of water content in egg white by pulse pressure-ultrasonic method and traditional method.

**Figure 3 foods-12-01477-f003:**
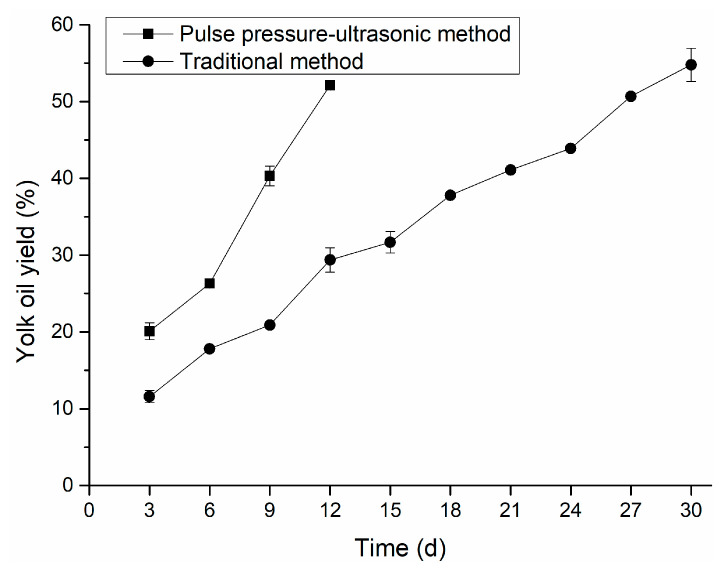
Comparison of yolk oil yield.

**Figure 4 foods-12-01477-f004:**
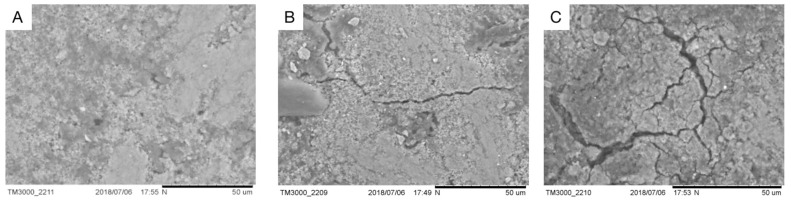
The surface of the shell with different preserved methods. (**A**) shows a fresh duck eggshell surface; (**B**) shows a conventional pickled eggshell surface; (**C**) shows a quickly pickled eggshell surface.

**Figure 5 foods-12-01477-f005:**
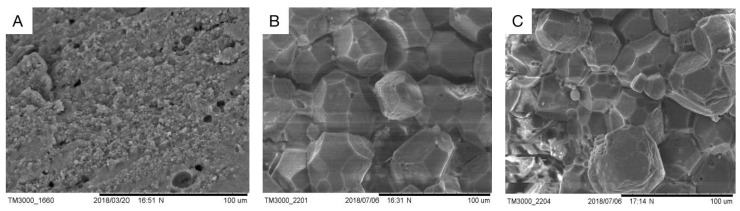
Egg yolk with different preservation methods. (**A**) illustrates the scanning electron microscope of a fresh duck egg yolk; (**B**) illustrates the scanning electron microscope of a conventionally pickled egg yolk; (**C**) illustrates the scanning electron microscope of a quickly pickled egg yolk.

**Figure 6 foods-12-01477-f006:**
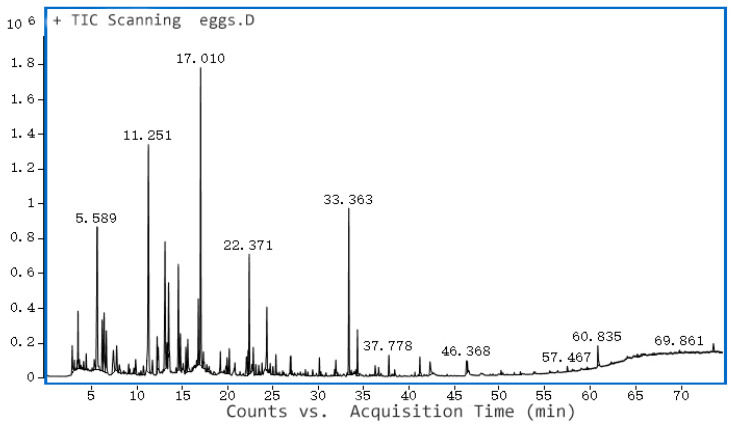
GC-MS of volatile compounds of fresh egg yolk.

**Figure 7 foods-12-01477-f007:**
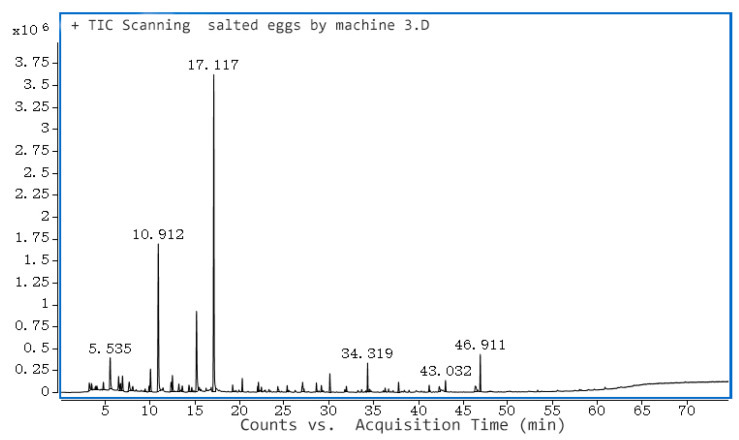
GC-MS of volatile compounds of quickly pickled egg yolk.

**Table 1 foods-12-01477-t001:** Texture profile analysis of egg white with salting time.

Time (d)	Hardness (N)	Springiness (mm)	Cohesiveness	Gumminess (g)	Chewiness (mJ)	Resilience
0	1038.40 ± 176.26 ^a^	0.80 ± 0.07 ^a^	0.70 ± 0.02 ^a^	712.82 ± 44.72 ^a^	570.65 ± 111.96 ^a^	0.27 ± 0.04 ^a^
3	907.53 ± 136.95 ^a^	0.91 ± 0.06 ^a^	0.69 ± 0.10 ^ab^	625.95 ± 15.64 ^a^	560.13 ± 97.78 ^a^	0.28 ± 0.02 ^a^
6	876.68 ± 84.01 ^ab^	0.87 ± 0.06 ^a^	0.68 ± 0.03 ^ab^	625.25 ± 37.79 ^a^	539.16 ± 79.89 ^a^	0.27 ± 0.02 ^a^
9	897.39 ± 56.23 ^ab^	0.80 ± 0.02 ^a^	0.69 ± 0.01 ^ab^	614.87 ± 23.69 ^a^	409.28 ± 47.75 ^b^	0.26 ± 0.01 ^a^
12	731.02 ± 87.46 ^bc^	0.81 ± 0.05 ^a^	0.67 ± 0.12 ^ab^	484.15 ± 54.69 ^b^	394.54 ± 50.15 ^b^	0.27 ± 0.04 ^a^
15	686.06 ± 97.86 ^c^	0.77 ± 0.18 ^a^	0.66 ± 0.02 ^b^	468.67 ± 25.63 ^b^	400.80 ± 78.44 ^b^	0.21 ± 0.01 ^b^

Values with different letters indicate a significant difference (*p* < 0.05).

**Table 2 foods-12-01477-t002:** Texture profile analysis of egg yolk with salting time.

Time (d)	Hardness (N)	Springiness (mm)	Cohesiveness	Gumminess (g)	Chewiness (mJ)	Resilience
0	672.79 ± 21.92 ^a^	0.83 ± 0.05 ^a^	0.76 ± 0.04 ^a^	507.53 ± 55.65 ^a^	413.14 ± 144.53 ^a^	0.38 ± 0.05 ^a^
3	628.58 ± 34.31 ^ab^	0.82 ± 0.01 ^a^	0.75 ± 0.01 ^a^	476.32 ± 28.20 ^ab^	334.21 ± 26.52 ^ab^	0.36 ± 0.0 ^ab^
6	606.39 ± 56.23 ^ab^	0.80 ± 0.02 ^a^	0.73 ± 0.03 ^a^	464.40 ± 45.75 ^ab^	409.28 ± 42.29 ^a^	0.35 ± 0.03 ^ab^
9	616.01 ± 64.75 ^ab^	0.79 ± 0.03 ^a^	0.63 ± 0.13 ^ab^	461.60 ± 58.61 ^ab^	367.96 ± 67.18 ^a^	0.35 ± 0.01 ^ab^
12	595.10 ± 29.67 ^ab^	0.78 ± 0.18 ^a^	0.66 ± 0.04 ^ab^	404.32 ± 23.67 ^ab^	291.33 ± 11.96 ^ab^	0.29 ± 0.05 ^b^
15	525.02 ± 46.15 ^b^	0.44 ± 0.06 ^b^	0.53 ± 0.02 ^b^	336.63 ± 46.51 ^b^	150.57 ± 57.56 ^b^	0.21 ± 0.04 ^c^

Values with different letters indicate a significant difference (*p* < 0.05).

**Table 3 foods-12-01477-t003:** Common pollutants in salted egg yolk flavor components.

No.	Name
1	Cyclohexane, methyl-
2	Cyclotrisiloxane, hexamethyl-
3	Cyclotetrasiloxane, octamethyl-
4	Cyclopentasiloxane, decamethyl-
5	Cyclohexasiloxane, dodecamethyl-

**Table 4 foods-12-01477-t004:** Analysis of volatile flavor components in salted egg yolk.

No.	Name	Relative Content (%)		
		Fresh Egg Yolk	Quick Curing	Conventional Pickling
1	Butanal, 3-methyl-	1.67	2.42	1.54
2	Ethanol	-	16.03	7.78
3	1-Hexanol	-	-	0.66
4	1-Octanol	-	-	0.21
5	2,3-Butanediol	-	-	1.84
6	1-Butanol, 3-methyl-	-	0.29	0.73
7	1-Octen-3-ol	1.29	2.11	8.39
8	1-Butanol	-	1.17	1.34
9	1-Pentanol	-	0.58	0.45
10	Benzenemethanol,. alpha.,.al	-	2.89	-
11	Hexanal	9.84	28.71	27.60
12	Pentanal	-	1.83	1.33
13	2-Heptanone	-	0.46	1.33
14	3-Octanone	-	-	1.13
15	Octanal	1.77	1.42	3.07
16	Benzaldehyde	0.68	0.91	2.64
17	Acetophenone	-	0.94	-
18	2-Nonanone	-	0.24	-
19	Nonanal	0.460	0.56	2.10
20	2-Octenal, (E)-	-	0.27	0.76
21	Acetoin	-	1.39	0.63
22	Toluene	0.48	9.04	1.36
23	Benzene, 1,3-bis (1,1-dimethylethyl)-	4.94	-	0.72
24	Phenol	-	-	0.39
25	1,3-Diazine	-	-	0.23
26	Furan, 2-pentyl-	-	0.77	3.50
27	Acetic acid, butyl ester	-	-	0.99
28	Butanoic acid, 3-hydroxy-,	-	-	0.26
29	Nonanoic acid	0.85	-	-
30	Acetic acid	-	0.26	0.34
31	Dimethyl sulfide	-	4.00	-
32	Oxime-, methoxy-phenyl-_	-	0.90	0.45
33	Pyrazine, 2,5-dimethyl-	-	-	0.63

## Data Availability

Data is contained within the article or Appendix A.

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
