# Peer review of "A Comparative Study of Pickled Salted Eggs by Positive and Negative Pressure-Ultrasonic Method"

_foods, 2023, doi:10.3390/foods12071477_

Round 1

Reviewer 1 Report

The manuscript entitled `A Comparative Study of Pickled Salted Eggs by Positive and Negative Pressure-Ultrasonic Method` shows interesting findings on effect of sonication on some quality properties of egg yolk and egg white during pickling process compared to conventional application.  However some improvements required for better representation.

In method section conventional method should be elaborated including the amount of salt content.

Similarly sonication process has lack of information. Author should give details about equipment, sonication device and pressure magnitude etc. There is no clear experimental design to follow up.

Author used egg white and protein for the same meaning, for better understanding one of them should be selected throughout the text.

Discussion section requires more comparison to novel research as well.

Therefore I recommend minor revisions. 

Author Response

Dear Editor:

We gratefully appreciate the editors and all reviewers for their time and making positive and constructive comments. These comments are all valuable and helpful for revising and improving our manuscript entitled “A Comparative Study of Pickled Salted Eggs by Positive and Negative Pressure-Ultrasonic Method” (ID: 2238518), as well as the important guiding significance to our researches.

We have studied comments carefully and have made correction which we hope meet with approval. Revised portion are marked in red in the revised manuscript. The summary of corrections and the responses to the reviewer’s comments are listed in the Revision Report.

Thank you and best wishes

Yours sincerely,

Xiao Chaogeng

E-mail: [email protected];

Reviewer 2 Report

1. Abstract is good but more findings can be included.

2. The Introduction The positive and negative pressure-ultrasonic method is not explained with support of previous studies.

3. Introduction is written nicely but lack the proper justification of present work, where is methodology adopted with reference to other studied or not.

4. In Section 2.3. “Positive and negative pressure-ultrasonic marinated salted eggs Duck eggs of similar size and without damage were selected, cleaned, and dried into pickling containers, poured with marinade, and sealed, and pickling parameters were set  (high-pressure amplitude of 120 kPa, low-pressure amplitude of -70 kPa, positive and 80 negative pressure time ratio of 16:24 min, ultrasonic effect in the first three days of pick- 81 ling, and daily action time of 30 min) for sampling tests”Author should elaborate the processs

5. Line  in Introduction section “The egg white samples were compressed to 70% of the original volume with the probe, 103 and the yolk samples were compressed to 80% of the original volume with the probe. The test speed was 1 mm s-1 with a dwell time of 5 s. Each sample was tested 10 times”

Author should elaborate the results in newton (N) force.

6. Section 3.2 Author has given Moisture Content sub heading, please explain the relevant of this heading in this manuscript. As the egg water content is given in the rest of the manuscript and in methodology the process of moisture content analysis is not given.Below reference can be studied and added in the manuscript for better understanding of moisture content.

Selected physico-mechanical characteristics of cryogenic and ambient ground turmeric. International Agrophysics. 28, 111-117.

7. Authors have given better figures and tables  for understands of viewers in various place but in table 1. How does the unit of Viscosity can be in Grams?

8. Section 3.2,3.3,3.5,3.6 and 3.7  are well explained and results are clear to understand but 3.4 needs improvement.

9.. As per the results findings the Conclusion given is not upto the mark, it should be  a complete gist of study in bullet points.

Author Response

(The authors gave the same response as above.)
